# Asymptomatic Carriage of *Listeria monocytogenes* by Animals and Humans and Its Impact on the Food Chain

**DOI:** 10.3390/foods11213472

**Published:** 2022-11-01

**Authors:** Dagmar Schoder, Claudia Guldimann, Erwin Märtlbauer

**Affiliations:** 1Department of Veterinary Public Health and Food Science, Institute of Food Safety, University of Veterinary Medicine, 1210 Vienna, Austria; 2Veterinarians without Borders Austria, 1210 Vienna, Austria; 3Department of Veterinary Sciences, Faculty of Veterinary Medicine, Institute of Food Safety and Analytics, Ludwig-Maximilians-University Munich, 85764 Oberschleißheim, Germany; 4Department of Veterinary Sciences, Faculty of Veterinary Medicine, Institute of Milk Hygiene, Ludwig-Maximilians-University Munich, 85764 Oberschleißheim, Germany

**Keywords:** domestic animals, ruminants, wildlife, human, crop, vegetable, environmental contamination, asymptomatic carriers, *Listeria monocytogenes*

## Abstract

Humans and animals can become asymptomatic carriers of *Listeria monocytogenes* and introduce the pathogen into their environment with their feces. In turn, this environmental contamination can become the source of food- and feed-borne illnesses in humans and animals, with the food production chain representing a continuum between the farm environment and human populations that are susceptible to listeriosis. Here, we update a review from 2012 and summarize the current knowledge on the asymptomatic carrier statuses in humans and animals. The data on fecal shedding by species with an impact on the food chain are summarized, and the ways by which asymptomatic carriers contribute to the risk of listeriosis in humans and animals are reviewed.

## 1. Introduction

Food-borne listeriosis caused by *Listeria monocytogenes* accounted for 1876 human cases in the EU in 2020. It is also the zoonosis with the highest case fatality rate of 10% in the EU [1]. Combined with the often severe neurological symptoms, this makes listeriosis a high priority for food safety efforts worldwide. *L. monocytogenes* has a broad host range in humans, as well as wild and domestic animals that typically become infected by the ingestion of food or feed that has been contaminated with *L. monocytogenes*.

Potential sources for *L. monocytogenes* in feed and food result from the ubiquitous presence of *L. monocytogenes* in the environment [2], fecal shedding by hosts and the ability of *L. monocytogenes* to establish itself in suitable niches in the farm or food-processing environment due to its capacity to adapt to a broad range of environmental stresses [3].

The food chain provides a direct link between the farm environment and human hosts. *L. monocytogenes* gains access to food production facilities through either raw materials of animal origin (meat and milk) via produce that are contaminated with *L. monocytogenes* from soil or feces or from other sources through a lack in hygiene management. A subset of strains of *L. monocytogenes* (e.g., clonal complex (CC) 9 or 121) have shown a higher propensity to persist in the food production environment, mainly through increased resistance against disinfectants such as quaternary ammonium compounds [4]. It is not uncommon for *L. monocytogenes* to persist in niches in food processing facilities for years or even decades [5].

Taken together, food intended for human consumption may become contaminated with *L. monocytogenes* at any level: (i) during primary production at the farm level, (ii) during processing or (iii) at the retail or (iv) consumer level due to insufficient hygiene measures during food handling. If *L. monocytogenes* is able to grow in food or feed matrices that are consumed without an inactivation step (e.g., heating), the basic conditions for an outbreak of listeriosis are met.

Infections of human or animal hosts result in clinical presentations that range from asymptomatic carriers to septicemia, encephalitis or abortions [6]. While the pathomechanisms in the host and the bacterial virulence factors in *L. monocytogenes* are well-understood, it remains largely unclear why some individuals become asymptomatic carriers. In some cases, truncated alleles of the gene *inIA* encoding surface protein internalin A were found in strains isolated from asymptomatic human carriers [7]. Truncated forms of the *inlA* gene were associated with a loss of virulence [8], which may lead to asymptomatic carriage, and were also overrepresented in isolates from food compared to clinical isolates [9], suggesting the potential exposure of consumers to these isolates. Additionally, a contribution of viable but nonculturable (VBNC) forms of *L. monocytogenes* [10] to asymptomatic carriage has been hypothesized [11,12].

Among other bacteria, *L. monocytogenes* has developed various mechanisms for switching from a vegetative to a metabolically inactive state. However, *L. monocytogenes* in the VBNC state represent a diagnostic challenge, because the majority of the current tests need at least one cultivation step, thus failing to detect nongrowing VBNC cells. Fortunately, in recent years, a variety of PCR and qPCR applications combined with DNA intercalating dyes have been established for detecting viable and VBNC cells [12].

Asymptomatic carriers present a major challenge to food safety (Figure 1). On the other hand, according to a recent study, the gut microbiota itself is a major line of defense against foodborne pathogens [13]. The authors point out that a specific microbiota signature is associated with the asymptomatic shedding of *L. monocytogenes*. They conclude that fecal carriage of this pathogen is a common phenomenon in healthy individuals and is very much influenced by the gut microbiota [13].

Efficient controls are in place to exclude animals that are obviously ill from milk or meat production. In contrast, the intermittent fecal shedding of asymptomatic carriers often remains invisible in humans and animals. [14]. With regard to farm animals, the fecal shedding of *L. monocytogenes* can lead to three contamination scenarios with potential implications for food safety: (i) It contributes to a higher load of *L. monocytogenes* in the immediate barn environment, increasing the risk of additional animals becoming infected. (ii) Manure from these animals may be used to fertilize fields, and runoff from farms may contaminate water sources, both risk factors for the contamination of feed and crops with *L. monocytogenes* [15,16]. (iii) Unrecognized carrier animals may lead to raw milk and meat contamination due to insufficient hygiene practices during milking or slaughtering. Finally, asymptomatic human colonization with *L. monocytogenes* may result in the direct contamination of food or the food processing environment due to insufficient hand hygiene.

The aim of this review is to update a 2012 review on the asymptomatic carrier statuses in different species [17] and to summarize the current knowledge on risk factors associated with fecal shedding in different species.

## 2. Domestic Animals as Asymptomatic Carriers

From studies on Listeria ecology, the asymptomatic carriage of *L. monocytogenes* seems to be evident worldwide, and most domestic animal species, including dogs and cats, can shed *L. monocytogenes* intermittently via feces (Table 1). In addition, the prevalence data showed significant counts of *L. monocytogenes* in tonsil samples from healthy domestic animals (Table 1). The role of household pets in spreading *L. monocytogenes* is not well-studied. To our knowledge, there is no documented clinical case of the transmission of *L. monocytogenes* from pets to humans. However, in recent years, it has become increasingly popular for dog and cat owners to feed their pets raw meat-based diets (RMBDs) instead of the more conventional dry or canned pet foods. A Dutch research team demonstrated that RMBDs may be a possible source of *L. monocytogenes* infection in pet animals and, if transmitted, pose a risk for human beings. They analyzed 35 commercial RMBDs from eight different brands. Alarmingly, *L. monocytogenes* was present in 54% of all tested samples [18].

Pigs can be important reservoirs for *L. monocytogenes* (Table 1), and in particular, younger animals are at risk for asymptomatic carriage. For example, the prevalence of *L. monocytogenes* in the tonsils of fattening pigs (22%) was significantly higher than in sows (6%) [19]. Hypervirulent clones of *L. monocytogenes* such as CC6 [9] were found in pig tonsils, and due to the presence of closely related isolates along the production chain, the cross-contamination or recontamination of meat from a specific source in the slaughterhouse seems to play an important role [20].

Housing conditions significantly influence the risk of *L. monocytogenes* detection in healthy pigs: According to Hellstrom et al. [21], there is a higher prevalence of *L. monocytogenes* in animals from organic production compared to conventional farms. The EU regulation on organic production (EU 2018/848) stipulates that pigs in organic production systems must have straw bedding and must have outdoor access. Additionally, pigs in organic production systems are typically housed in larger groups [21], all of which may contribute to a higher exposure of the animals to *L. monocytogenes* from the environment or from other animals within the same group. On the other hand, it cannot be denied that pigs of intensive indoor farming are often exposed to prolonged social, environmental and metabolic stress [22], which may also enhance the shedding of *L. monocytogenes*.

A highly variable prevalence has been found in fecal samples of healthy dairy cattle ranging from ±1.9% in individual animals to ≥46% of beef herds [23]. Antibody titers to specific *L. monocytogenes* virulence proteins, such as listeriolysin O and internalin A, were demonstrated in 11% of 1652 healthy dairy cows in Switzerland, suggesting that contact with *L. monocytogenes* is relatively frequent in this animal species [24]. A large-scale longitudinal study conducted to monitor *Listeria* spp. in dairy farms during three consecutive seasons in Spain showed that the prevalence of *L. monocytogenes* was affected by season and age: a higher prevalence was observed during the winter in cattle, and cows in their second lactation had the highest probability of *L. monocytogenes* fecal shedding [25].

In all likelihood, the fecal shedding of *L. monocytogenes* by cattle depends on extraneous factors, including feedstuff contamination and season. Shedding appears to be directly associated with feeding practices. A higher prevalence of *L. monocytogenes* in feces has occurred in farms with contaminated feed. Generally, *Listeria* spp. and *L. monocytogenes* prevalence were higher during the indoor season compared to the pasture season [26,27]. *L. monocytogenes* shedding by cows on a study farm was (i) dependent on the subtype of *L. monocytogenes*, (ii) highly associated with silage contamination and (iii) related to animal stress [28]. Poor-quality silage with fermentation defects can have pH values that are permissive to the growth of *L. monocytogenes* and act as a major risk factor for listeriosis in ruminants [29]. This may also explain the seasonal shedding patterns of *L. monocytogenes* by ruminants [14,30,31]. Additionally, for sheep and goats, the likelihood of the isolation of *L. monocytogenes* was three to seven times higher in farms that relied on silage feeding compared to those without [32]. Finally, it is interesting to note that *L. monocytogenes* clonal complex 1 is the most prevalent clonal group associated with human listeriosis and is strongly associated with cattle and dairy products [33].

Given the general proclivity of *L. monocytogenes* for most vertebrates, the special association of *L. monocytogenes* with ruminants may be a specific host adaptation that reflects the unique conditions in the pre-fermentative ruminant fore-stomach. It is the voluminous rumen that may favor the rapid multiplication of *L. monocytogenes* at a pH between 6.5 and 7.2 and at body temperatures between 38.0 and 40.5 °C before confrontation with the acidic environment of the abomasum. This hypothesis is supported by findings that brief, and the low-level fecal excretion of *L. monocytogenes* in sheep is concomitant with a transitory asymptomatic infection after translocation from the gastrointestinal tract (GIT), with the rumen digesta serving as a reservoir. In this study, the asymptomatic carriage of *L. monocytogenes* in sheep was not simply a case of passive passage of the bacteria but was associated with transitory multiplication in the rumen, depending on the dose of *L. monocytogenes* ingested and the age of the animal [34].

Poultry, turkeys, ducks and geese can asymptomatically carry *L. monocytogenes* [17]. Recently, carcass rinses and cloacal swabs were reported to be positive at a level of 11 and 1.3%, respectively [35,36]. In addition, there are numerous reports about contamination rates in poultry production establishments and poultry meat and meat products. Stress, such as transport, is plausibly one important factor that exacerbates shedding and thus contributes to the contamination of production lines.

In summary, healthy domestic animals can be asymptomatic carriers of *L. monocytogenes.* While the prevalence of fecal shedding tends to be low, husbandry practices involving silage feeding, as well as stressors associated with housing conditions, group sizes and transport are risk factors that can increase fecal shedding.

## 3. Carriage of *L. monocytogenes* by Wild Animals

Typically, studies on wildlife shedding *L. monocytogenes* provide no metadata on the health status of the animals, either because it was not evaluated, because animal droppings were sampled in the absence of the animal or because the symptoms of listeriosis were difficult to spot or unknown in a species. Additionally, catch and release studies may be biased towards animals with an impaired health status that may render them more likely to be caught. This makes a classification as “asymptomatic” carriers in wild animals problematic. However, for the purpose of this review, we assume that wild animals that are fecal shedders, symptomatic or not, contribute to the distribution of *L. monocytogenes* in and between environments and should therefore be considered in food safety risk assessments.

A variety of birds, including pheasants, pigeons, gulls, crows, rooks and sparrows, have been shown to be asymptomatic carriers of *L. monocytogenes* (Table 2). A comprehensive prevalence study in Japan looked at fecal or intestinal samples from 996 birds across 18 species and found *Listeria* spp. in 13.4% of all samples, most commonly in samples from crows [48]. Additionally, the fecal presence of *L. monocytogenes* was shown in 33% of urban rooks [49]. According to Hellstrom et al. [50], feces from wild birds (mostly from gulls, pigeons and sparrows) collected in Finland exhibited an overall *L. monocytogenes* prevalence of 36%. Pulsotypes obtained from the birds were often similar to those collected from the food chain, suggesting a possible role of birds in the spread of *L. monocytogenes* strains that are relevant in the context of human infections.

The carriage of *L. monocytogenes* in wildlife is not confined to wild birds. Table 2 shows that *Listeria* spp., including *L. monocytogenes*, have been isolated from a broad variety of mammals (e.g., deer, rodents and wild boars) and also other vertebrates such as reptiles. A Japanese study [48] that included fecal or intestinal samples from 623 wild mammals from eleven species identified *Listeria* spp. in 38 (6.1%) of the tested animals. The highest number of *Listeria* spp. isolates (16/38) were from monkeys, which resulted in a prevalence of 20.0% (16/80) in the monkey samples. A similar study conducted in Canada analyzed 268 fecal samples from a variety of animals, 112 of which were from wildlife, including deer, moose, otters and raccoons. Among these, 35 samples were positive for *L. monocytogenes* (29%) [51]. In samples of 45 red deer and 49 wild boars hunted in Austria and Germany during 2011/12, a total of 19 (42.2%) red deer were positive for *L. monocytogenes*, as were 4 (18.2%) out of 22 pooled feed samples and 12 (24.5%) boars [52]. In several samples, *L. monocytogenes* was isolated from the tonsils and ruminal or cecal contents without its presence in feces, implying that game can carry *L. monocytogenes* even if it is not detectable in their feces. The highest counts for *L. monocytogenes* were found in the rumen of deer and in the tonsils of boars. Pulsed-field gel electrophoresis showed a wide variety of strains, but the serotypes were predominantly 1/2a and 4b. A Polish study examining free-living carnivores as potential sources of infection [53] isolated *L. monocytogenes* from approximately 5% of animals, which included red foxes, beech martens and raccoons. A full set of intact virulence genes was present in 35% of the isolates; the remainder contained varying numbers and configurations of the genes.

Ready-to-eat fish and seafood products—in particular, cold smoked salmon—are frequent sources of human listeriosis [1]. The majority of cases are likely a consequence of post-harvest contamination by *L. monocytogenes* strains that persist in food processing facilities [65]. For farmed fish, factors such as water pollution, agricultural runoff and seagull feces are important contributing factors to the presence of *L. monocytogenes* in the fish and the farm environment [66]. Another alternative is the fish, such as salmon from wild catch. According to a recent Norwegian study, freshly slaughtered salmon contaminated with *L. monocytogenes* was a likely source for the introduction and subsequent persistence in a salmon processing plant [67]. However, fish themselves do not seem to be very susceptible to *L. monocytogenes*. After a gavage of *L. monocytogenes* into the stomachs of live salmon, they were readily cleared without pathologic changes to the animals within three days [66], and rainbow trout held in fish farms where *L. monocytogenes* was detected in the water were rarely positive for *L. monocytogenes* [68]. A recent study [61] demonstrated an increase in the *L. monocytogenes* contamination level in tilapia from capture (1.2%) to the domestic market (5.8%). Taken together, these data suggest that fish may become transient asymptomatic carriers of *L. monocytogenes* after exposure but are not likely to be long-term spreaders of the pathogen.

Reptiles [64], insects [69] and even protozoa [70] may also harbor *L. monocytogenes*. The ongoing trend to keep reptiles, such as snakes, turtles and geckos, as exotic pets should not go unmentioned. In Europe alone, it is estimated that more than 11 million reptiles were kept as pets in 2021 [71]. Further studies are required to assess the possible risk of infection for reptile keepers.

Lately, invading Spanish slugs (*Arion vulgaris*) have been implicated as vectors for *L. monocytogenes* [72]. Of the pooled samples of 710 slugs, 43% were positive, and 16% of them had mean counts of 405 CFU/g of slug tissue. Of 62 slugs cultured, 11% had a positive surface or mucus. Additionally, when the slugs were fed with *L. monocytogenes*, they shed viable bacteria in their feces for up to 22 days. Recently, ants were found to harbor *L. monocytogenes* sporadically, and their potential to transmit pathogenic microorganisms from contaminated environments to food has been demonstrated [69].

Overall, these data show that a wide range of vertebrates, including reptiles, birds and mammals, as well as some invertebrates, can act as carriers of *L. monocytogenes* and contribute to its spread between habitats through asymptomatic carriage.

## 4. Asymptomatic Carriage of *L. monocytogenes* in Humans

The fecal transmission of *L. monocytogenes* is not only confined to domestic and wild animals. Humans have been shown to shed *L. monocytogenes* intermittently, with the prevalence of fecal shedding in healthy individuals determined by cultures typically ranging below 5% (Table 3); for older studies, see [73].

Interestingly, although low levels of carriage were found in Austria [74] and the USA [75] for healthy people, a later study in Austria compared feces from three individuals sampled over a three-year period. They found that ten (1.2%) out of 868 samples proved positive for *L. monocytogenes*, all of which were serotypes 1/2a and 1/2b. A closer analysis revealed that there were five independent asymptomatic exposures to the bacterium, corresponding to an average of two exposures per person per year [76]. According to the scientific opinion of the European Food Safety Authority on *L. monocytogenes* contamination of RTE foods and the risk for human health in the EU, there is an increasing number of clinical cases for the over 75 years of age group and female age group between 25 and 44 years old. Quantitative modeling demonstrated that more than 90% of invasive listeriosis is caused by the ingestion of RTE food containing > 2000 CFU/g and that one-third of cases are due to growth of the organism in the consumer phase [77].

Underlying medical conditions may also be a predisposing factor for the asymptomatic carriage of *L. monocytogenes* in humans—for instance, in patients on renal dialysis who received the H_2_ receptor antagonist antacid ranitidine [78]. On the other hand, the fecal prevalence of *Listeria* spp. or *L. monocytogenes* was the same between HIV-infected pregnant women receiving antiretrovirals and uninfected controls [79]. Pregnancy itself does not seem to affect the rate of human asymptomatic carriage. It was shown that 51 women in their 10–16th weeks of pregnancy had a fecal carriage rate of only 2% [80]. In comparison, the same fecal carriage was confirmed for 3.4% out of 59 nonpregnant controls. Moreover, when the fecal carriage rates in pregnant women with listeriosis were compared with matched, nonpregnant controls following an outbreak in Los Angeles, similar carriage rates were found [81]. A recent study [82] indicated that dysbiosis of breast milk microbiota may result in an increased relative abundance of *L. monocytogenes* in the milk of the mothers of children showing severe acute malnutrition (SAM).

A culture-independent approach based on molecular methods detected *L. monocytogenes* in 173/3338 (5.2%) human microbiome datasets on MG-RAST (16S sequencing) and in 90/900 (10%) stool samples from healthy individuals using PCR [13]. The interpretation of these data should bear in mind that DNA-based detection methods do not differentiate between live and dead organisms. The same study also showed a correlation between specific gut microbiota and the presence of *L. monocytogenes*. Interestingly, a study in mice indicated that aging may cause significant dysbiosis of the commensal microbiota, resulting in increased *L. monocytogenes* colonization of the gut [88]. Additionally, occupational groups encountering animals, feces and meat and those who undergo work-related exposure to the bacterium are anticipated to be at an increased risk of asymptomatic infection. For example, the cumulative prevalence of *L. monocytogenes* in hand swabs from farm workers and hand and clothes swabs from abattoir workers was 16% and 6%, respectively [83,86], which is higher than the average prevalence typically found in fecal samples from healthy people (Table 3).

In order to grasp the extent of *L. monocytogenes* exposure in the wider human community, a European Union-wide baseline survey was carried out in 2010 and 2011. All in all, 13,088 food samples were examined for the presence of *L. monocytogenes*. The prevalence across the entire European Union in fish samples was 10.4%, while, for meat and cheese samples, the prevalence were 2.07% and 0.47%, respectively [89].

Wagner et al. [90] sampled ready-to-eat foods in Austria. Out of 946 food samples collected from food retailers in Vienna, 124 (13.1%) and 45 (4.8%) tested positive for *Listeria* spp. and *L. monocytogenes*, respectively. Products showing the highest contamination were fish and seafood (19.4%), followed by raw meat sausages (6.3%), soft cheese (5.5%) and cooked meats (4.5%). The samples were also collected from households in the same region, and 5.6% and 1.7% out of 640 foodstuffs tested positive for *Listeria* spp. and *L. monocytogenes*, respectively. Alarmingly, the same isolates from the latter products could be detected from pooled fecal samples of household members, suggesting that even low-level contaminated foods (<100 CFU/g) may result in fecal shedding.

## 5. The Impact of Asymptomatic Carriers on the Presence of *L. monocytogenes* on Farms and in the Food Processing Environment

As discussed above, asymptomatic fecal shedding of *L. monocytogenes* by farm animals contributes to an increased presence of the pathogen in the farm environment with an associated risk to food and feed safety. A systems approach to food safety therefore should include a thorough analysis of the ecology of *L. monocytogenes* in the agricultural environment, and the identification and elimination of farm reservoirs for *L. monocytogenes* is a prerequisite for the implementation of farm-specific pathogen reduction programs [91]. Table 4 summarizes the recent studies that were performed in this context.

We can conclude that asymptomatic fecal shedding by farm animals is linked to diet, particularly to silage [92]. The incidence of *L. monocytogenes* silages was reported to range from 2.5% (clamp silage) to 22.2% (large bales) and to be even higher (44%) in moldy samples [93]. *L. monocytogenes* is thought to initially access silage from the contamination of raw grass via soil. Insufficient acidification of the silage caused by inadequate fermentation then allows the growth of *L. monocytogenes* to levels that can cause disease. Surprisingly, *L. monocytogenes* was rarely detected on grass and vegetables prior to processing [92], which may reflect the low numbers of bacteria needed as an initial contamination. However, once plant material is contaminated, the bacteria can survive for weeks, with implications for feed safety if the grass is contaminated and food safety when the crops are contaminated [94]. Importantly, recent studies in the USA demonstrated that the use of surface water for irrigation could be a major source of contamination [95,96,97]. *L. monocytogenes* was found in up to 27% of the samples of pond water [98] and in up to up to 99% in the waste water of stabilization ponds in the arctic region of Canada [99]. In addition, the ability of *L. monocytogenes* to enter the VBNC state may contribute to adaptation, persistence and transmission between different ecological niches [11].

Besides silage, *L. monocytogenes* was regularly isolated from samples obtained from feed bunks, water troughs and bedding [47,100], which is consistent with its ubiquitous presence in soil and subsequent spread through feed and animals. Most interestingly, recent studies indicated that dairy farms may favor the selection of hypervirulent *L. monocytogenes* clones, which can then enter the food chain [4,25].

Taken together, the persistence of *L. monocytogenes* in the ruminant farm environment may be supported by a cycle of ingestion of *L. monocytogenes* with contaminated feed, multiplication in animal hosts and subsequent fecal contamination of the environment [101].

Although pigs seldom develop clinical listeriosis, pork products have consistently been linked to human infection [1,102,103]. Slaughter and processing environment contaminations have been traced back to healthy carrier pigs [104]. As for in–out or empty and clean finishing pig facilities, when the duration of the empty period prior to the introduction of growing pigs was less than one day in the fattening section, the risk of *L. monocytogenes* contamination was significantly increased [105]. This same group also proposed that wet feeding is a risk factor for *L. monocytogenes* colonization of a finishing batch, likely because of feed residue layers and biofilm formation within the pipes and valves. The prevalence of fecal shedding of *L. monocytogenes* in healthy swine generally increases from the farm to food manufacturing plants [91]. However, the main source for *L. monocytogenes* contamination in food appears to be at the slaughter and processing steps, where the bacterium can survive for very long periods [104].

Taken together, the asymptomatic shedding of humans and animals, as well as *L. monocytogenes* persistence in the farm environment, present a risk to animal and human health. Since *L. monocytogenes* may access food production facilities from these primary sources, preventative strategies at this level of the food production chain should focus on a high standard of feed and animal hygiene, sanitary milk production and good farming practices. Poor hygiene on farms, such as inattention to boot cleaning, hand washing, failure to wear protective clothing and indifference to silage quality, increases the risk of animals becoming colonized with *L. monocytogenes*, including the downstream risk to the human consumer.

The colonization of food processing equipment and facilities can originate from raw food sources or introduction by poor hygiene practices or fomites [114]. Persistent strains of *L. monocytogenes* isolated from the food processing environment show enhanced adherence with short contact times, promoting survival and possibly initiating the establishment of a strain as a “house strain” in a food processing plant [115]. Sodium chloride, which is often used in food production, induces autoaggregation and increases *L. monocytogenes* adhesion to plastic [114]. The same authors found that persistent strains might have a lower virulence potential than clinical strains. Others have also observed that *L. monocytogenes* strains responsible for persistent contamination differ from sporadic strains, but there does not seem to be any specific evolutional lineage of persistent strains [19]. Disturbingly, this lower virulence may change following exposure to disinfectants [116,117].

Asymptomatic fecal shedding of *L. monocytogenes* by cows can lead to the entry of *L. monocytogenes* into dairy processing plants via contaminated raw milk and result in persistence as disinfectant-tolerant biofilms on surfaces and the subsequent contamination of processed products [101]. While *L. monocytogenes* is killed by short-term, high-temperature pasteurization, it can survive and thrive in post-pasteurization processing environments and thereby recontaminate dairy products [118,119,120]. The unique growth and survival properties of *L. monocytogenes* and its ability to adhere to surfaces contribute to the difficulty of eliminating it entirely [121].

In a poultry processing facility in Northern Ireland, a particular genotype of *L. monocytogenes*, considered to originate from incoming birds and prevalent in the raw meat processing area, was found to be widespread on food contact surfaces, floors and drains [122]. One year later, the strains isolated from cooked poultry products and the cooked poultry-processing environment contained only that genotype, plus one other, common to both raw and cooked meat areas. This highlights the potential for persistent strains to cross-contaminate processed foods in the same facility.

## 6. Future Implications

Since it is evident that animals and humans can persistently carry *L. monocytogenes* and thereby act as a source for the contamination of processed food, control measures must begin on the farm and include humans, animals and the environment in a one-health approach. An awareness of asymptomatic carriage should inform hygiene regulations with respect to the animal and food handlers at all stages of food production. In the farm context, proactive farm hygiene practices to lower the bacterial burden on crops and in animal feed can reduce the root causes of *L. monocytogenes* access to animal and human hosts. In particular, sewage handling and irrigation techniques for crops should take into account the possibility of spreading *L. monocytogenes* to growing plants. Attention to feed hygiene and to correct fermentation during silage production helps interrupt the cycle between the shedding of *L. monocytogenes* from asymptomatic ruminant carriers to grass via manure and the subsequent colonization of more animals from contaminated feed.

Meaningful preventive measures include the adequate compartmentalizing of the raw food processing steps, the critical rethinking of the need for silage feeding, avoiding irrigation close to the harvest, the scrupulous cleanliness of food contact surfaces and equipment, the strict personal hygiene of food handlers and regular monitoring for the persistent colonization of the food processing environment with *L. monocytogenes*.

## Figures and Tables

**Figure 1 foods-11-03472-f001:**
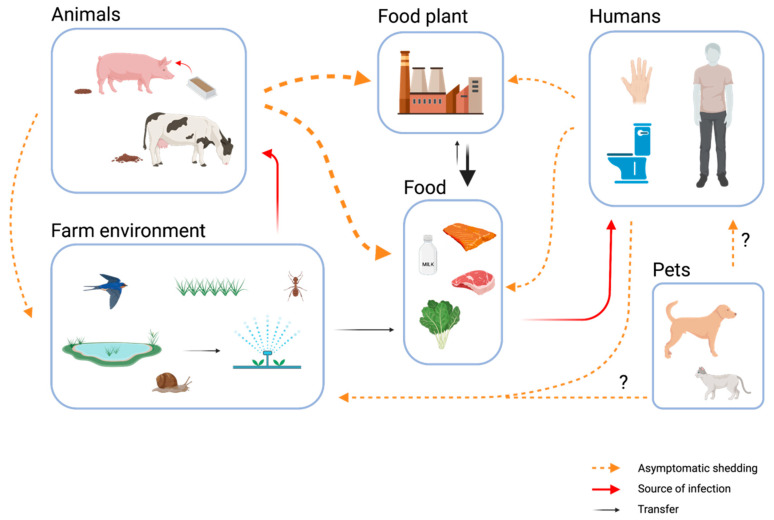
Role of asymptomatic fecal shedding of *L. monocytogenes* by humans and animals in the spread between habitats with a focus on food safety. Bold arrows indicate the most relevant contamination pathways in food production chains in Europe today—in particular, along the farm–food plant–food continuum. Figure was created with BioRender.com.

**Table 1 foods-11-03472-t001:** Isolation of *Listeria monocytogenes* from healthy domestic animals.

Country	Animal	Target	Sample (*n*)	Positive (%)	Ref.
Austria	Sheep/Goat	*L.* spp.	Feces (53)	42.6	[32]
		*L. monocytogenes*		13	
Egypt	Cattle	*L. monocytogenes*	Feces (660)	6.8	[37]
			Milk (660)	5.9	
Finland	Chicken	*L monocytogenes*	Cloacal swabs (457)	1.3	[36]
Germany	Cattle	*L. monocytogenes*	Feces (138)	33	[38]
	Sheep		(100)	8	
	Hens		(100)	8	
	Pigs		(34)	5.9	
	Horses		(400)	4.8	
	Dogs		(300)	1.3	
	Cats		(275)	0.4	
Germany	Pigs	*L. monocytogenes*	Tonsils (430)	1.6	[39]
		*L. innocua*		1.2	
Italy	Pigs	*L. monocytogenes*	Salivary gland, lymph nodes, tonsils (189)	13.2	[40]
Japan	Cattle	*L. monocytogenes*	Feces (1705)	1.9	[41]
	Pigs			0.6	
	Dogs			0.9	
	Rats			6.5	
Japan	Cattle	*L. monocytogenes*	Feces (1738)	6	[42]
Jordan	Cattle	*L. monocytogenes*	Feces (610)	1.5	[43]
Qatar	Camel	*L. monocytogenes*	Feces (50)	4	[44]
Slovenia	Cows	*L. monocytogenes*	Feces (540)	18.2	[45] *
	Calves		(511)	8.4	
Spain	Cattle (beef)	*L. monocytogenes*	Feces (301)	42.3	[23]
	Cattle (dairy)			46.3	
	Sheep			23.5	
Spain	Cattle	*L. monocytogenes*	Feces (953)	4.3	[25]
	Sheep		Feces (483)	5.8	
	Goat		Feces (333)	0.3	
Switzerland	Cattle	Ab to LO	Serum (1652)	11	[24]
		and IA **			
Taiwan	Chicken	*L. monocytogenes*	Carcass rinse (246)	11.4	[35]
USA	Cattle	*L. monocytogenes*	Feces (825)	31	[14] ***
USA	Cattle	*L. monocytogenes*	Feces (528)	20.2	[31] ****
USA	Broiler	*L. monocytogenes*	Feces (555)	14.9	[46]
USA (Central NY State)	Cattle	*L. monocytogenes*	Milk (1412)	13	[47]
			Udder swab (1408)	19	
			Feces (1414)	43	

* Fecal samples were collected from cows and calves on 20 family dairy farms in 2-week intervals for a period of 1 year. ** Antibodies to listeriolysin O and internalin A. *** Twenty-five fecal samples were collected daily for two 2-week periods and one 5-day period. **** A case–control study involving 24 case farms with at least one recent case of listeriosis and 28 matched control farms with no listeriosis cases was conducted to study the transmission and ecology of *Listeria monocytogenes* on farms.

**Table 2 foods-11-03472-t002:** Isolation of *Listeria monocytogenes* from healthy wild animals.

Country	Animal	Target	Sample (n)	Positive (%)	Ref.
Austria/ Germany	Red deer	*L. monocytogenes*	samples* (45)	42	[52]
	Wild boar		(49)	25	
Bulgaria	Birds (*Riparia riparia, Motacilla flava*)	*L. monocytogenes*	Feces (673)	0.6	[54]
Canada	Geese	*L.* spp.	Feces (495)	9.5	[55]
		*L. monocytogenes*		4.0	
Canada	Gulls (*Laurus delawarensis*)	*L. monocytogenes*	Cloacal swabs (264)	9.5	[56]
China	Rodents	*L. m*	Feces (702)	0.3	[57]
		*L. ivanovii*		3.7	
		*L. innocua*		6.7	
China	Rodents	*L.* spp.	Feces (341)	9	[58]
		*L. monocytogenes*		3.2	
		*L. innocua*		2.9	
Finland	Birds	*L. monocytogenes*	Feces (212)	36	[50]
Finland/Norway	Reindeer	*L. monocytogenes*	Feces (470)	3.2	[59]
France	Rooks	*L. monocytogenes*	Feces (112)	33	[49]
		*L. innocua*		24	
		*L. seeligeri*		8	
Germany	Pigeons	*L. monocytogenes*	Feces (350)	0.8	[60]
		*L. innocua*		2.3	
		*L. seeligeri*		0.6	
Japan	Crows	*L. monocytogenes*	Feces (301)	1.7	[48]
		*L. innocua*		43	
Kenya	Nile tilapia	*L. monocytogenes*	Muscle (167)	1.2	[61]
Poland	Red deer	*L. monocytogenes*	Feces (120)	1.75	[62]
Poland	Red fox, beech marten, racoon	*L. monocytogenes*	Rectal swab (286)	3.5	[53]
			(65)	6.1	
			(70)	4.3	
Switzerland	Wild boars	*L. monocytogenes*	Tonsils (153)	17	[63]
			Feces (153)	1	
USA (central New York)	Reptiles	*L. monocytogenes*	Feces (17)	12	[64]
	Mammals		(64)	8	
	Birds		(242)	4.5	

* Tonsils and content of the rumen or the stomach, liver, intestinal lymph nodes, cecum content and feces.

**Table 3 foods-11-03472-t003:** Humans as carriers of *Listeria* spp.

Country	Target	Sample (n)	Positive (%)	Reference
Austria	*L. monocytogenes*	Feces, healthy people (505)	0.2	[74]
Brazil	*L. monocytogenes*	Feces, pregnant women (213)	2.4	[79]
	*L.* spp.		7.5	
Egypt	*L. monocytogenes*	Hand swabs, farm workers (100)	16	[83]
	*L. innocua*		2	
France	*hly* gene	Feces (900)	10	[13]
Germany	*L. monocytogenes*	Feces, patients with diarrhea (1000)	0.6	[84]
	*L. innocua*		1.7	
Germany	*L. monocytogenes*	Feces, healthy people (2000)	0.8	[84]
	*L. innocua*		2	
Iran	*L. monocytogenes*	Feces (80)	7.5	[85]
		Vaginal swabs (80 samples from women with at least two abortions)	11.3	
Senegal	*L. monocytogenes*	Breast milk, mothers of SAM children (120)	100	[82]
		Breast milk, mothers of healthy children (32)	37.5	
Turkey	*L. monocytogenes*	Hand and clothes swabs, abattoir workers (70)	5.7	[86]
Turkey	*L. monocytogenes*	Feces (1061)	0.9	[87]
UK	*L. monocytogenes*	Feces, patients with gastroenteritis (171)	1.8	[78]
USA	*L. monocytogenes*	Feces (827)	0.12	[75]

**Table 4 foods-11-03472-t004:** *Listeria* spp. isolated from the farm and environment.

Country	Target	Sample (n)	Positive (%)	Ref.
Austria	*L.* spp.	Working Boots (53)	51	[32]
		Floor (53)	39.3	
	*L. monocytogenes*	Working Boots (53)	15.7	
		Floor (53)	7.9	
Canada	*L. monocytogenes*	Irrigation water (223)	10.3	[106]
Canada (Arctic region)	*L. monocytogenes*	Wastewater stabilization ponds (109)	99	[99]
Denmark	*L. monocytogenes*	Abattoir poultry (3080)	8.0	[107]
Egypt	*L. monocytogenes*	Water (36)	8.3	[37]
		Silage (36)	27.8	
		Manure (36)	19.4	
		Soil (36)	8.3	
		Milking equipment (432)	6.9	
Germany	*L. monocytogenes*	Slaughterhouse (environment and equipment, 77)	0.9	[20]
Iran	*L. monocytogenes*	Water (180)	16.7	[85]
Iran	*L. monocytogenes*	Iranian currency (108)	0.93	[108]
Jordan	*L. monocytogenes*	Bulk tank milk (305)	7.5	[43]
New Zealand	*L. monocytogenes*	Bulk tank milk (400)	4.0	[109] *
South Africa	*L. monocytogenes*	Roof-harvested rainwater (264)	22	[110]
Taiwan	*L. monocytogenes*	Abattoir environment (246)	0	[35]
USA	*L. monocytogenes*	Soil (555)	15.3	[46]
USA	*L.* spp.**	Stream water (196)	28	[95]
	*L. monocytogenes*		10	
USA (Colorado, wilderness area)	*L. monocytogenes*	Soil, water, sediment, surface soil and wildlife fecal samples (572)	0.23	[111]
	*L.* spp.		1.5	
USA (Idaho)	*L. monocytogenes*	Dairy wastewater ponds (30)	6.7	[112]
USA (New York State)	*L. monocytogenes*	Water (132)	48	[113]
	*L. innocua*		10.6	
	*L. monocytogenes*	Feces (77)	29	
	*L. innocua*		8	
USA (NYS)	*L.* spp.	Spinach field soil (1092)	12	[96]
	*L. monocytogenes*		7.8	
USA (NYS)	*L.* spp.	Pond/river water used for irrigation (9)	44	[97]
	*L. monocytogenes*		22	
USA (Virginia)	*L. monocytogenes*	irrigation water		[98]
		pond (48)	27.1	
		well (48)	4.2	

* Survey from November 2011 to August 2012 during which 25-mL milk samples were collected five times from each of 80 randomly selected dairy farms and tested for the presence of *L. monocytogenes*. ** Excluding *L. monocytogenes*.

## Data Availability

Not applicable.

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
