# Peer review of "Asymptomatic Carriage of *Listeria monocytogenes* by Animals and Humans and Its Impact on the Food Chain"

_foods, 2022, doi:10.3390/foods11213472_

Round 1
Reviewer 1 Report
The review entitled: "Asymptomatic carriage of Listeria monocytogenes by animals and humans and its impact on the food chain" is characterized by the up-to-date information on the possible source of asymptomatic animal and human carriers to transmit this pathogen in the food processing environment. The article is well written. However, some information should be added to provide the full range of data.
Pg. 1, Ln 13-first mentioning- write the full name L. monocytogenes
Pg. 3- 2. Domestic animals as asymptomatic carriers
Although there is no documented clinical case of transmission of L. monocytogenes from pets to humans this should not be disregarded since dogs and cats are mostly asymptomatic carriers but can shed L. monocytogenes to the environment. The role of pet food as aa a potential source of these bacteria in pets should be mentioned. Although feeding raw meat-based diets (RMBDs) to companion animals has been popular a 54% of the RMBD products contained Listeria monocytogenes and 43% of other Listeria spp, according to a study performed in Dutch (https://doi.org/10.1136/vr.104535).
Pg. 8, Ln 225- Reptiles [60],-this should be considered also from the view of keeping reptiles as exotic pets. Other wild animals that are sometimes kept as pets could be asymptomatic carriers of L. monocytogenes.
Author Response
The review entitled: "Asymptomatic carriage of Listeria monocytogenes by animals and humans and its impact on the food chain" is characterized by the up-to-date information on the possible source of asymptomatic animal and human carriers to transmit this pathogen in the food processing environment. The article is well written. However, some information should be added to provide the full range of data.
Pg. 1, Ln 13-first mentioning- write the full name L. monocytogenes
- has been changed;
Pg. 3- 2. Domestic animals as asymptomatic carriers
Although there is no documented clinical case of transmission of L. monocytogenes from pets to humans this should not be disregarded since dogs and cats are mostly asymptomatic carriers but can shed L. monocytogenes to the environment. The role of pet food as aa a potential source of these bacteria in pets should be mentioned. Although feeding raw meat-based diets (RMBDs) to companion animals has been popular a 54% of the RMBD products contained Listeria monocytogenes and 43% of other Listeria spp, according to a study performed in Dutch (https://doi.org/10.1136/vr.104535).
- thank you for this hint; information has been included;
Pg. 8, Ln 225- Reptiles [60],-this should be considered also from the view of keeping reptiles as exotic pets. Other wild animals that are sometimes kept as pets could be asymptomatic carriers of L. monocytogenes.
- information has been included (including current statistical data on reptile husbandry in Europe)
Submission Date
13 September 2022
Date of this review
19 Sep 2022 18:03:25
Reviewer 2 Report
Very interesting and clear manuscript.
Some suggestions/modifications :
Line 70 remove : very unlikely situation (people ill with listeriosis are not able to work due to infection gravity).
L103 to modulate since stress in intensive housing conditions may also enhance shedding
L119-140 are very interesting, the role of cattle in CC1 shedding and transmission may be cited. (Moura et al., Sci. Adv. 7, eabj9805 (2021) 1 December 2021)
L219-222 to modulate since in smoked salmon and trout production, farm and raw material contamination are well known (XIX.2 WORKSHOP MÉTODOS RÁPIDOS Y AUTOMATIZACIÓN
EN MICROBIOLOGÍA ALIMENTARIA –memorial DYCFung–Barcelona (2021 ; http://jornades.uab.cat/workshopmrama)
L244-245 to link to data on prevalence and risk exposure
L276-283 add the 2010 european baseline study survey: Anonymous (2013). Analysis of the baseline survey on the prevalence of Listeria monocytogenes in certain ready-to-eat (RTE) foods in the EU, 2010-2011. Part A: Listeria monocytogenes prevalence estimates. EFSA J. 11, 3241.
L286 to remove: not appropriate since it is asymptomatic carriage and not infections
L292 environment
L398 link between vbnc and infection never has been proved, remove or modulate
Author Response
Very interesting and clear manuscript.
Some suggestions/modifications :
Line 70 remove : very unlikely situation (people ill with listeriosis are not able to work due to infection gravity).
- has been removed
L103 to modulate since stress in intensive housing conditions may also enhance shedding
- information about intensive housing conditions has been included;
L119-140 are very interesting, the role of cattle in CC1 shedding and transmission may be cited. (Moura et al., Sci. Adv. 7, eabj9805 (2021) 1 December 2021)
- has been included;
L219-222 to modulate since in smoked salmon and trout production, farm and raw material contamination are well known (XIX.2 WORKSHOP MÉTODOS RÁPIDOS Y AUTOMATIZACIÓN
EN MICROBIOLOGÍA ALIMENTARIA –memorial DYCFung–Barcelona (2021 ; http://jornades.uab.cat/workshopmrama)
- has been modulated accordingly (information covers also wild catch and farmed fish)
L244-245 to link to data on prevalence and risk exposure
- has been included
L276-283 add the 2010 european baseline study survey: Anonymous (2013). Analysis of the baseline survey on the prevalence of Listeria monocytogenes in certain ready-to-eat (RTE) foods in the EU, 2010-2011. Part A: Listeria monocytogenes prevalence estimates. EFSA J. 11, 3241.
- has been included
L286 to remove: not appropriate since it is asymptomatic carriage and not infections
- has been removed
L292 environment
- has been corrected
L398 link between vbnc and infection never has been proved, remove or modulate
- has been removed
Submission Date
13 September 2022
Date of this review
26 Sep 2022 09:48:23
Reviewer 3 Report
Major revisions:
Lack of information about the reason of asymptomatic carriage of Listeria monocytogenes fenomenom.
Lack of description of diagnostic methods to detect L. monocytogenes, especially in VBNC state.
Line 54: truncated alleles of surface protein internalin A (encoded by inlA) > truncated alleles of gene inlA encoding surface protein internalin A L. monocytogenes
Line 55: Truncated forms of inlA were > Truncated forms of inlA gene were
Line 58: Additionally, a contribution of viable .... This is based on comparisons with Mycobacterium tuberculosis, where entry in a VBNC state in latent tuberculosis has been shown > That's a bad comparison Mycobacterium tuberculosis, where entry in a VBNC state still could be detected with some modification using BACTEC or normal, but prolonged culture with Stonebrink TBor Lowenstein-Jensen Medium. Moreover, PCR for detection Listeria and other bacteria in VBNC state is well known technique.
Author Response
Lack of information about the reason of asymptomatic carriage of Listeria monocytogenes fenomenom.
- Phenomenon “asymptomatic carriage” has been described; see line 76-81;
Lack of description of diagnostic methods to detect L. monocytogenes, especially in VBNC state.
- Information has been included (both, diagnostic limits and new, modern molecular biological methods for the detection of VBNC L. monocytogenes cells); see line 69-75;
Line 54: truncated alleles of surface protein internalin A (encoded by inlA) > truncated alleles of gene inlA encoding surface protein internalin A L. monocytogenes
- has been corrected
Line 55: Truncated forms of inlA were > Truncated forms of inlA gene were
- has been corrected
Line 58: Additionally, a contribution of viable .... This is based on comparisons with Mycobacterium tuberculosis, where entry in a VBNC state in latent tuberculosis has been shown > That's a bad comparison Mycobacterium tuberculosis, where entry in a VBNC state still could be detected with some modification using BACTEC or normal, but prolonged culture with Stonebrink TBor Lowenstein-Jensen Medium. Moreover, PCR for detection Listeria and other bacteria in VBNC state is well known technique.
- Thank you very much for this hint; sentence has been removed;
Submission Date
13 September 2022
Date of this review
23 Sep 2022 14:47:31
Round 2
Reviewer 3 Report
The amended article is clearer.
I only have one observation:
Line 69 - 75: a variety of PCR and qPCR applications combined with DNA intercalating dyes have been established for detecting viable and VBNC cells - intercalating dyes e.g. SYBR, but not only. It should be added also real-time PCR technique using probes (e.g. TaqMan probe).